# Outcomes of COVID-19 in Pregnant Women: A Retrospective Analysis of 300 Cases in Jordan

**DOI:** 10.3390/healthcare12212113

**Published:** 2024-10-23

**Authors:** Hamza Hasan Suliman Al-Amoosh, Rasmieh Al-Amer, Aysheh Hasan Alamoush, Fatima Alquran, Taghreed Mohammad Atallah Aldajeh, Taysier Ahmad Al Rahamneh, Amer Gharaibeh, Amira Mohammed Ali, Maher Maaita, Tamara Darwish

**Affiliations:** 1Faculty of Medicine, Gynecology and Obstetrics Department, Hashemite University, Zarqa 13133, Jordan; hamzaho@hu.edu.jo; 2Nursing Faculty, Mental Health Department, Yarmouk University, Irbid 21163, Jordan; 3Faculty of Nursing, Adult Health Nursing Department, Applied Science Private University, Amman 11937, Jordan; a_alamoush@asu.edu.jo; 4Gynecology and Obstetrics Department, King Hussein Medical Center, Amman 11733, Jordan; fatima_quran@yahoo.com.au (F.A.); mahmoudhindawi55@yahoo.com (T.M.A.A.); 5Gynecology and Obstetrics Department, Royal Medical Services, Amman 11855, Jordan; tayseerabadi1@gmail.com (T.A.A.R.); agharaibeh@hotmail.com (A.G.); mmaaita@hotmail.com (M.M.); 6Department of Psychiatric Nursing and Mental Health, Faculty of Nursing, Alexandria University, Alexandria 21526, Egypt; amira.mohali@alexu.edu.eg; 7King’s College Hospitals, London SE5 9RS, UK; t.darwish1@nhs.net

**Keywords:** COVID-19, maternal outcome, neonatal outcomes, demographics, comorbidities

## Abstract

Background: The impact of COVID-19 on pregnancy remains a critical area of research, with growing evidence suggesting that maternal infection, particularly in the third trimester, may lead to significant complications Aims: The primary aim was to investigate the maternal and neonatal outcome of pregnant Jordanian women with COVID-19. The secondary aim included exploring demographics, obstetrics characteristics, and comorbidities among these women. Methods: A retrospective comprehensive review of the records of 300 cases of pregnant women with COVID-19, who were treated between November 2020 and April 2021 at Queen Alia Military Hospital (a main referral center for patients with COVID-19) in Jordan. All cases were confirmed by the rapid antigen test (RAT) + long polymerase chain reaction (PCR) test used to detect SARS-CoV-2 by amplifying viral RNA from patient samples. Women infected with COVID-19 were categorized into four groups according to the RCOG guidelines for COVID-19 infection in pregnancy: asymptomatic, mild, moderate, and severe cases. All cases were managed following the Royal College of Obstetricians and Gynecologists protocol for COVID-19 in pregnancy. Data extracted from patient’s records included demographic information, COVID-19 clinical manifestations, obstetric history, diagnostic findings, treatment plans, comorbidities, gestational age at diagnosis, treatment protocols, and maternal and neonatal outcomes. Results: The mean age was 29.7 years; 98.3% were nonsmokers; 8% had previous miscarriages, and 67.3% had the infection in the third trimester. Iron deficiency anemia affected 30.3%, while 18.3% had comorbidities, mainly hypothyroidism. Most women were asymptomatic 61.7%, but 33% had respiratory symptoms, 4.7% needed intensive care unit (ICU) admission, and 2.7% resulted in maternal deaths. First-trimester and second-trimester miscarriages were recorded in 2.67% and 3.67% of cases, respectively, while preterm labor occurred in 3.0% of pregnancies. Additionally, age and hospitalization duration had a positive correlation with the neonatal outcomes (r = 0.349, *p* < 0.01), (r = 0.376, *p* < 0.01), respectively. Furthermore, COVID-19 presentation and treatment options demonstrated a strong positive correlation (*p*-value <0.01). On the other hand, maternal death had a strong negative correlation with poor neonatal outcomes (r = −0.776, *p* < 0.01). Conclusion: The study showed that COVID-19 in pregnant women, particularly in the third trimester, is associated with significant neonatal complications, with age, hospitalization duration, and COVID-19 severity strongly impacting outcomes.

## 1. Introduction

The global healthcare landscape underwent unprecedented challenges during the COVID-19 pandemic [1]. Among the myriad affected populations, pregnant women stand out as a unique demographic, requiring special consideration due to physiological adaptations and potential vulnerabilities caused by pregnancy [2,3]. Extensive research has been conducted to understand the clinical manifestations of COVID-19. However, there remains a discernible gap in the literature: a comprehensive case series examining maternal and neonatal outcomes in a substantial cohort of 300 COVID-19-positive pregnancies has yet to be conducted.

This study is situated within a broader research context that has urged specialized investigations into COVID-19′s impact on pregnant populations, recognizing the need for a nuanced understanding of the intersection between pregnancy and novel viral infection [4]. While diverse clinical presentations of COVID-19 have been explored [5,6], the specific implications for pregnant women remain a crucial area of investigation.

The physiological changes inherent in pregnancy may influence the course of COVID-19, emphasizing the necessity for targeted exploration in this unique population [3]. Additionally, understanding the potential vertical transmission of COVID-19 during pregnancy is paramount, as highlighted by previous research [7]. Comorbidities have emerged as a critical factor influencing the severity of COVID-19 [8]. Likewise, the gestational age at which COVID-19 is diagnosed has been identified as a crucial variable impacting outcomes, underlining the importance of a nuanced approach to the management of pregnant women during the pandemic [9].

The outcomes of neonates born to mothers with COVID-19 have also been explored in-depth, shedding light on potential ramifications for the offspring of affected pregnancies [10]. However, a comprehensive case series incorporating variables such as hospitalization durations, age distributions, smoking habits, gravidity, parity, obstetric history, comorbidities, pregnancy complications, COVID-19 clinical manifestations, oxygen saturation levels, disease severity, treatment protocols, and maternal and neonatal outcomes within a substantial cohort of 300 cases is conspicuously absent in the current literature.

This study aimed to fill this gap by examining both maternal and neonatal outcomes in a case series of 300 COVID-19-positive pregnancies. The findings are anticipated to contribute substantially to the existing body of knowledge, providing valuable insights for healthcare practitioners, policymakers, and researchers involved in managing pregnant women during the COVID-19 pandemic.

Hence, this study answers the following research questions:1.What are the demographics, obstetrics characteristics, and comorbidities among Jordanian pregnant women with COVID-19?2.What are the COVID-19-related characteristics among pregnant women in Jordan?3.What are the maternal outcomes among Jordanian pregnant women with COVID-19?4.What are the obstetrics and neonatal outcomes among pregnant women diagnosed with COVID-19 and their offspring?5.What factors are correlated with COVID-19 among pregnant women diagnosed with COVID-19?

## 2. Methods

### 2.1. Design, Setting, and Sampling

This research adopted a retrospective case series design, meticulously analyzing the maternal and neonatal outcomes of COVID-19-positive pregnancies. During the COVID-19 pandemic, Queen Alia Military Hospital (QAMH) was assigned by the Royal Medical Services (RMS) to be the main referral center for all COVID-19 patients including pregnant women. An exemplary multidisciplinary team that includes obstetricians, anesthetists, internal medicine specialists, respiratory specialists, chest physiotherapists, and intensivists was assigned to manage these patients. All pregnant women with laboratory-confirmed diagnoses of COVID-19 between October 2020 and April 2021 were included in this study.

### 2.2. Confirmation and Evaluation of COVID-19 Cases

Each suspected case was confirmed by the rapid antigen test (RAT) + long PCR [11]. All COVID-19 patients were assessed and evaluated by an obstetrician and an internist. Proper history-taking includes the onset, timing, and severity of symptoms, contacts with infected patients, vaccinations, and the presence of any comorbidities. A comprehensive examination was carried out, including body mass index (BMI), pulse rate, temperature, respiratory rate, blood pressure, and oxygen (O_2_) saturation on air room and on exertion. Pregnant women infected with COVID-19 were sent to the gynecology ward (isolation section), to be re-evaluated by an obstetrician where full obstetric history was taken and bedside ultrasonography done. Laboratory tests including complete blood count (CBC), kidney function test (KFT), liver function test (LFT), prothrombin time/international normalized ratio (PT/INR), protein fragment present in the blood after a blood clot dissolves, used to rule out thrombotic disorders (D-Dimer), C-reactive protein (CRP), lactate dehydrogenase (LDH), and ferritin were undertaken for all patients. Chest X-ray and high-resolution computed tomography (CT) (HRCT) were performed with abdominal shields after a patient-informed consent. The chest X-ray was used for initial evaluation, while the HRCT was conducted based on specific criteria, such as the need for detailed imaging due to abnormal findings on the X-ray or clinical symptoms indicating more complex lung conditions. Following this comprehensive assessment, women infected with COVID-19 were categorized into four groups according to the RCOG guidelines for COVID-19 infection in pregnancy [12]:

Asymptomatic: Positive PCR without symptoms;

Mild: Any signs and symptoms of COVID-19 and no respiratory disease;

Moderate: Lower respiratory tract disease with O_2_ saturation equal to or more than 94%;

Severe: O_2_ saturation less than 94%, respiratory rate (RR) more than 30, X-rays showing lung infiltrates more than 50%.

Each of the above groups was managed differently and according to the clinical guidelines established by RCOG [12].

### 2.3. Criteria for Admission and Outpatient Management in COVID-19 Pregnancies

The decision whether to admit the pregnant women or treat them as an outpatient was made on a case-by-case basis depending on the following criteria: (a) clinical presentation; (b) HRCT findings; (c) need for supportive care; (d) potential risk factors for severe illness; and (e) the ability of the patient to self-isolate at home.

There were four management protocols according to the RCOG guidelines for COVID-19 infection in pregnancy:A.Conservative management for the asymptomatic groupB.Zocin/piperacillin/tazobactam 500 mg, an antibiotic given once daily (OD) for five days; vitamin C (VITC), 500 mg tablet twice daily (BD); vitamin D (VITD), 1000 u once daily (OD); and zinc, 25 mg once daily (OD).C.Admission, rocephin/ceftriaxone, an antibiotic given at 2 g once daily (OD); zocin/piperacillin/tazobactam, 500 mg/an antibiotic given once daily (OD) for five days; decadron/dexamethasone, a corticosteroid given intravenously (IV) at 6 mg once daily (OD); innohep/tinzaparin 4500 IU, low molecular weight heparin (anticoagulant) given at 4500 IU once daily (OD); VIT C, 500 mg tablet twice daily (BD); VIT D, 1000 u once daily (OD); and zinc, 25 mg once daily (OD); famotidine 40 mg OD/famotidine, an acid-reducing medication given at 40 mg once daily (OD).D.Admission, rocephin 2 g OD, zocin 500 mg OD, decadron IV 6 mg OD, innohep 4500 IU OD, VIT C 500 mg TAB BD, VIT D 1000 U OD, zinc 25 mg OD, famotidine 40 mg OD plus/minus the following according to the severity and duration of the disease: antiviral treatment (remdesivir or sancovir), tocilizumab (actemra), and/or hemoperfusion:

Those categorized as severe were treated with either C or D protocol, depending on the severity of the disease, duration of the disease, and gestational age.

### 2.4. Ethical Approval

This study was approved by the Research Committee of the School of Medicine at Hashemite University. The Ethics Committee waived the need for patient consent. To illustrate, the researcher applied for a waiver of patient consent because of the retrospective design utilized in this study. The waiver was granted based on the following criteria: (a) the research posed minimal or no risk to the women, (b) the data were anonymized before analysis to protect patients’ confidentiality, (c) obtaining individual consent was impractical due to the historical nature of the data, and (d) the study holds significant potential to contribute to understanding the maternal and neonatal outcomes of pregnant Jordanian women with COVID-19 and to explore the factors correlated with the health-related outcomes for both the mothers and their infants.

### 2.5. Data Collection

After receiving the ethical clearance from the School of Medicine Ethics Committee in Hashemite University, we approached the manager of the relevant department at Queen Alia Military Hospital and requested their assistance in facilitating the process of data collection after explaining the aims and inclusion criteria of the study. Afterwards, we were granted permission to extract the relevant data from patients’ medical records from diverse healthcare facilities covering the period from October 2020 to April 2021. The cases were collected consecutively by extracting data from the records of all eligible patients over the study period, ensuring no bias in case selection. The inclusion criteria were pregnant women diagnosed with COVID-19, confirmed through PCR testing. The exclusion criteria were women who suffer from chronic obstructive pulmonary disease (COPD) and lung cancer.

The authors defined specific data points to be extracted from each patient record, including demographic information, COVID-19 presentations, obstetric history, diagnostic findings, treatment plans, comorbidities, gestational age at diagnosis, treatment protocols, and maternal and neonatal outcomes. All patient records were anonymized to protect patient privacy and comply with ethical standards. Following that, the first researcher reviewed a random sample of extracted records to ensure that data was accurately pulled according to predefined criteria. Furthermore, three researchers from the team were engaged to cross-verify the extracted data for accuracy. Any discrepancies were identified and corrected. To further ensure consistency and accuracy, the extracted data were compiled into a standardized format for analysis.

### 2.6. Measures

The current study utilized five data sheets to extract the required data. These sheets were developed in advance by the researchers based on existing literature and systematically captured data, including demographics, obstetric history, COVID-19-related data, maternal outcomes, and neonatal outcomes. The following paragraphs provide a brief discussion of each data sheet.

The demographic sheet includes two variables: age and smoking habits. The obstetric history sheet comprises ten variables, including obstetric history, previous mode of delivery, gestational age at diagnosis, comorbidities, and pregnancy complications. The COVID-19-related data sheet covers four variables: clinical presentation, O_2_ saturation, disease severity, and treatment protocol. The maternal outcomes sheet consists of the following variables: duration of hospitalization, treatment options (inpatient vs. outpatient), ICU admission, and maternal death. Lastly, the pregnancy and neonatal outcomes sheet contains two variables: mode of delivery and neonatal outcomes.

### 2.7. Data Analysis

For data analysis and management, the study used the Statistical Package for Social Sciences (IMB SPSS) version 25 (Chicago, IL, USA). Once data collection was complete, the data of this study were exported from an Excel sheet to an SPSS sheet. Afterward, the data set was checked for any deviant cases or outliers. Descriptive statistics, including frequencies and percentages, were used to summarize categorical variables. Continuous variables were presented as means with standard deviations or medians with interquartile ranges, depending on the distribution. The correlation coefficient was used to examine the strength of the relationship between the study-related variables. The statistical significance of the correlation was assessed using a *p*-value, with a *p*-value less than 0.05 considered significant.

## 3. Results

The mean age of study participants was 29.7 years with the majority falling in the category between 21–30 years (57%). The results show that 295 women (98.3%) were non-smokers, and 24 women (8.0%) had had at least one previous miscarriage. Meaning, most women 269 (89.7%) never experienced a miscarriage. In relation to the timing of infection, 59 women (19.7%) were in their second trimester of pregnancy. Regarding previous modes of birth, 47 women (15.7%) had at least one previous cesarean delivery. In addition, 91 women (30.3%) were found to have iron deficiency anemia based on laboratory findings. More details are depicted in Table 1.

### 3.1. Distribution of Comorbidities Among Patients with COVID-19 in Pregnancy Who Reported Comobrbitis

Among the 55 participants who reported comorbidities, 31.1% had hypothyroidism, 21.8% had diabetes mellitus, 9.3% had hypertension, 9.3% had asthma, and 10.9% had morbid obesity. Other comorbidities included hypertension and diabetes mellitus (3.8%), ovarian cysts (3.3%), thalassemia carrier (3.3%), epilepsy (1.6%), hepatitis A (1.6%), and hydronephrosis (1.6%), more details are depicted in Figure 1.

### 3.2. COVID-19 Related Characteristics Among Pregnant Women in Jordan

Table 2 shows that 182 (61.7%) of patients were asymptomatic. On the other hand, 99 (33%) patients presented with respiratory symptoms, while 7 (2.3%) presented with gastrointestinal (GI) symptoms only. As shown in Table 2, 15 (5.0%) and 12 (4.0%) had O_2_ saturations between 90%-94% and 80%-89%, respectively. The severity of the disease was classified into four categories, as previously mentioned: asymptomatic, mild, moderate, and severe. Among the total sample, 53.3% were asymptomatic (160 cases), 21% had mild symptoms (63 cases), 6.7% presented with moderate symptoms (50 cases), and 9.0% exhibited severe symptoms (27 cases).

Furthermore, Table 2 shows that depending on the severity of the symptoms, patients received different treatment protocols; 160 (53.3%) of the participants received protocol A, 62 (20.7%) received protocol B, 34 (11.3%) received protocol C, and 44 (14.7%) received protocol D.

### 3.3. Maternal Outcomes Among Jordanian Pregnant Women with COVID-19

Table 3 shows that 168 (56%) women were treated as outpatients and had a good prognosis, while 109 (36.3%) women were inpatients with a good prognosis as well. However, 14 (4.7%) women needed ICU admission at some point during their stay. Unfortunately, 8 (2.7%) maternal deaths occurred, and 1 patient refused admission.

### 3.4. Obstetrics and Neonatal Outcomes Among Pregnant Women Diagnosed with COVID-19

Table 4 shows obstetrics outcomes. In total, 35.7% of study cases had normal vaginal births, while 50.3% had cesarean sections. Among those who delivered surgically, 32.0% had elective cesarean sections, and 18.3% had emergency cesarean sections for variable indications. First-trimester and second-trimester miscarriages were recorded in 8 (2.67%) and 11 (3.67%) cases, respectively, while preterm labour occurred in 3.0% of pregnancies. Neonatal outcomes were categorized as follows: 87.3% born alive, 4.0% stillbirths, and 6.3% miscarriages. More details are depicted in Table 3.

### 3.5. COVID-19 Correlated Factors Among Pregnant Women with the Infection

Table 5 illustrates that age was positively correlated with neonatal outcomes (r = 0.349, *p* < 0.01), hospitalization duration (r = 0.376, *p* < 0.01), and ICU admission (r_pb = 0.560, *p* < 0.05). Additionally, COVID-19 clinical manifestations and treatment settings demonstrated a strong positive correlation (r_pb = 0.430, *p* < 0.01), (r_pb = 0.610, *p* < 0.01), respectively, and these levels of correlation suggest a significant relationship with the outcome. On the other hand, maternal death had a strong negative correlation, indicating that higher instances of maternal death were associated with poor neonatal outcomes (r_pb = −0.776, *p* < 0.01).

## 4. Discussion

The outcomes of 300 COVID-19-positive pregnancies at a central referral unit in the heart of Amman, Jordan presents a nuanced picture of the impact of the virus on maternal and neonatal health. In parallel with existing literature, this provides a comprehensive understanding of the broader trends while highlighting the unique aspects of our study [13,14].

Demographically, our study mirrors the age distribution observed by Ullah et al. 2020 [15] and Akhtar et al. 2020 [16], with a significant proportion of 56.7% falling in the 21–30 age group. This emphasizes the vulnerability of younger individuals to COVID-19 during pregnancy and necessitates targeted preventive measures.

Regarding clinical presentation, our data mirrored that yielded globally. Most of our patients were asymptomatic. The most common presenting symptoms among our patients were respiratory (33%). A prospective study published in the UK also had the same outcome and is in line with other reports from the United States and China [17,18]. These consistent observations suggest that the presentation of COVID-19 is comparable among pregnant women in different locations.

A noteworthy deviation from the existing literature is the higher prevalence of hypothyroidism in our cohort (5.7%), which contrasts with the healthier profiles reported by Dararei et al. 2019 [19]. This indicates the need for further exploration into the interplay between pre-existing conditions and COVID-19 outcomes during pregnancy [20,21]. Of note also, our study has yielded no significant association between severe COVID-19 infection and pre-eclampsia unlike numerous other studies [22]. However, such discrepancy could be explained by the fact that severe COVID-19 infection could mimic pre-eclampsia clinically.

Pregnancy outcomes among women with COVID-19 infection have luckily been thoroughly investigated by different researchers including a published meta-analysis, making comparable data outcomes readily available. Our research outcomes corroborate the association between COVID-19 and adverse obstetrics outcomes such as surgical delivery [23], preterm labor (3.0%), and miscarriage (6.3%).

Cesarean delivery rate is increased among women with COVID-19 infection. A cohort study conducted in China by Yang et al. confirmed higher rates of surgical delivery among women with COVID-19 infection [24]. This finding which goes in line with ours, has also been highlighted in a meta-analysis by Jafari et al. [25]. The increased rate of cesarean delivery could be attributed to worsening maternal health caused by the infection itself or its associated morbidities [18]. Another study conducted by Prabhu et al. contributed to this rise in fever during labor caused by the virus yet treated as chorioamnionitis based on clinical presentation [26].

The incidence of maternal death yielded in our study (2.7%) goes in line with reports published by the CDC stating a 70% higher incidence of death among pregnant women with COVID-19 compared with non-pregnant women [27]. We believe, however, that our outcome should be approached with caution, and warrants further investigation into factors contributing to this variation from global outcomes as the absolute risk of maternal death globally remains low. For instance, in the UK the risk of death associated with COVID-19 is 2.2 per 100,000 maternities. This discrepancy could possibly related to regional disparities or differences in healthcare infrastructure [28,29,30].

Preterm birth, mostly iatrogenic, has been linked to COVID-19 infection due to numerous factors attributed to the viral infection itself and those related to fetal indications. This rate, however, seemed to differ according to geographic location. European studies reported the highest rates of preterm births (19%), followed by Chinese studies (17%), while American studies demonstrated the lowest rates (12%) [31]. This disparity could be attributed to differences in obstetrics guidelines adhered to by different regions including ours.

The rate of miscarriage among our patients was significant at 6.3%. One retrospective study conducted in France yielded (2%) rate of miscarriage [32]. In another study conducted in Montreal, Quebec, women with asymptomatic COVID-19 infection were not found to have higher rates of first-trimester miscarriage [33]. We understand that the role of COVID-19 in miscarriage can be difficult to study due to the complexity of miscarriage itself and the already high rates of first-trimester miscarriage among women due to fetal abnormalities.

In terms of hospitalization duration, our findings suggest a median stay of 1.8 days, aligning with the shorter durations of previous analyses of COVID-19-positive pregnancies. This trend indicates a potential shift towards more outpatient management [15,34]. With reference to neonatal outcomes, our findings also align with previous studies indicating good outcomes where 87.3% were born alive while 4.0% were stillbirths. This variation may stem from differences in healthcare practices or regional disparities in our study population [30,35]. A systematic review conducted by Trevisanuto et al. concluded that reported no to mild symptoms of the disease among neonates even those born to mothers with severe illnesses [36]. This is not the only meta-analysis to report such outcomes, another published meta-analysis concluded that most neonates born to mothers with confirmed COVID-19 did not show any clinical signs of the infection [37].

In synthesizing these results, it is evident that the impact of COVID-19 on maternal and neonatal health is complex and influenced by various factors. While our study ratifies broader trends, the observed variations underscore the need for localized and context-specific approaches to managing pregnancies during novel viral pandemics.

## 5. Limitations and Strengths of the Study

Although we believe that this study provided important empirical data related to COVID-19 among pregnant women in a developing country, the results of this study should be seen within its limitations. This study used retrospective case series design which may limit definite causal interpretation between the study-related variables and outcomes. For example, in many cases, detailed information on the precise cause of the abortion or death may not have been comprehensively documented. Also, data were collected from existing records, which may lack some details or consistency. Moreover, the data of the current study were collected between October 2020 and April 2021; therefore, the changes in COVID-19 treatment protocols could have resulted in temporal bias which could have imposed some implications in the interpretation of the results. However, this study also has significant strengths. The results of this study are robust as the researchers undertook cross-validation several times. Besides, the current study report is based on a large database which helps minimize errors. The study addresses a critical gap in the literature by focusing on both maternal and neonatal outcomes in the context of COVID-19, providing a thorough understanding of how the virus affects pregnancy outcomes, especially in the third trimester. In addition to that, this study identified significant risk factors including, age, hospitalization duration, COVID-19 presentation, treatment options, and maternal/neonatal outcomes, providing valuable information into the factors influencing outcomes. This contributes to understanding risk factors and can help in guiding clinical decision-making.

## 6. Conclusions

The findings of this study highlight the difficulties in managing COVID-19 during pregnancy and stress the need for tailored healthcare strategies and better preparation for future pandemics. For example, health policymakers are invited to use the findings of his study in managing infectious disease in pregnant women and, in preparing the healthcare system on how to deal with future pandemics. Lessons learned from managing COVID-19-positive pregnancy could be used to enhance protocols for managing other respiratory infections during pregnancy. Importantly, highlighting the long-term maternal and neonatal outcomes in relation to COVID-19-positive pregnancies would have an impact on any health issue that might arise years after the initial infection. Furthermore, this study adds to the literature on how pandemics affect pregnancy in a developing country and will be used in comparative studies in the future.

## Figures and Tables

**Figure 1 healthcare-12-02113-f001:**
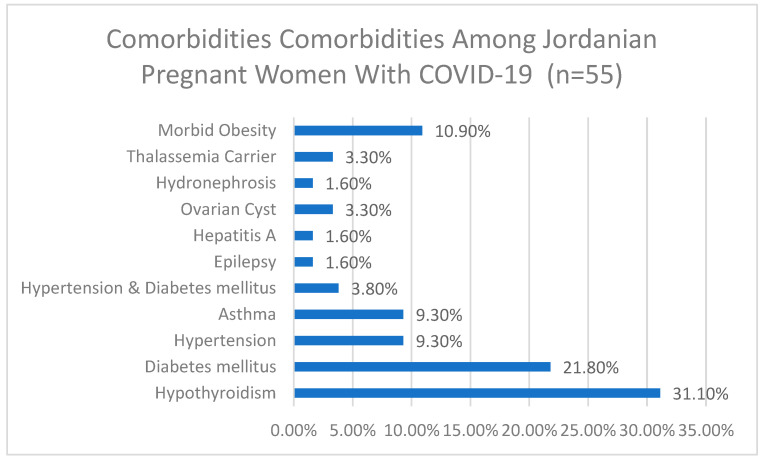
Comorbidities among Jordanian pregnant women with COVID-19 (n = 55).

**Table 1 healthcare-12-02113-t001:** The demographics and obstetrics data among pregnant women with COVID-19 (n = 300).

Age Group Median and (IQR); 21 (10)	Number (%)
17–20	11 (3.6)
21–30	170 (56.7)
31–40	108 (36.1)
40<	11 (3.6)
Smoking	
Yes	5 (1.7)
No	295 (98.3)
Obstraric History	
Gravidity	
1–3 times	182 (60.7)
4–5 times	79 (26.3)
>5 times	39 (13.0)
Number of Miscarriages Median and (IQR); 0 (2)	
No miscarriages	269 (89.7)
1–3 miscarriages	24 (8.0)
≥4	7 (2.3)
Previous Cesarean Section	
None	244 (81.3)
1–3 times	47 (15.7)
>3 times	9 (3.0)
Gestational Age Median and (IQR); 3 (1)	
First trimester	27 (9.0)
Second trimester	59 (19.7)
Third trimester	202 (67.3)
Missing	12 (4.0)
Pregnancy Complications	
None	153 (51.0)
Anemia	91 (30.3)
Gestational diabetes mellitus	5 (1.7)
Anemia and vaginal bleeding	5 (1.7)
Vaginal bleeding	5 (1.7)
Gestational hypertension	5 (1.7)
Anemia and post-partum hemorrhage	4 (1.3)
Post-partum hemorrhage	4 (1.3)
Others	73 (9.6)

**Table 2 healthcare-12-02113-t002:** The COVID-19-related data among pregnant women in Jordan (n = 300).

COVID-19 Presentation	Number (%)
No respiratory/GI symptoms	182 (60.7)
Respiratory symptoms	99 (33.0)
Gastrointestinal and respiratory symptoms	10 (3.3)
Gastrointestinal symptoms	7 (2.3)
Encephalitis	1 (0.3)
Gastrointestinal and vaginal bleeding	1 (0.3)
Oxygen Saturation %	
>95%	269 (89.7)
90–94%	15 (5.0)
80–89%	12 (4.0)
70–79%	4 (1.3)
Disease Severity	
Asymptomatic	160 (53.3)
Mild	63 (21.0)
Moderate	50 (16.7)
Severe	27 (9.0)
Treatment Protocol	
Protocol A	160 (53.3)
Protocol B	62 (20.7)
Protocol C	34 (11.3)
Protocol D	44 (14.7)

**Table 3 healthcare-12-02113-t003:** Maternal outcome among pregnant women with COVID-19 in Jordan (n = 300).

Hospital Stay Days	Number (%)	Mean (* SD); [Range]:
		**1.8, (2.48); [0–15]**
<5 days	271 (90.3)	
5–10 days	17 (5.7)	
>10 days	12 (4.0)	
Treatment settings		
Inpatient	109 (36.3)	
Outpatient	168 (56.0)	
ICU * Admission		
Yes	14 (4.7)	
No	286 (95)	
Maternal Death		
Total deaths	8 (2.7%)	
Refused admission	1 (0.3)	

* SD, Standard deviation; * ICU, intensive care unit.

**Table 4 healthcare-12-02113-t004:** Obstetrics and neonatal outcomes among Jordanian women with COVID-19 (n = 300).

Pregnancy Outcomes	Number (%)
Miscarriage	19 (6.3)
Preterm labour	9 (3.0)
Normal vaginal delivery	107 (35.7)
Induction of labor	6 (2.0)
Elective lower segment cesarean section	96 (32.0)
Emergency lower segment cesarean section	55 (18.3)
Intrauterine Fetal Death	5 (1.7)
Vacuum delivery	2 (0.7)
Termination of pregnancy due to severe COVID-19 infection	1 (0.3)
Neonatal Outcomes	
Alive baby	262 (87.3)
Stillborn	12 (4.0)
Miscarriage	19 (6.3)
Twin 1 alive 1 dead	2 (0.7)
No available data	1 (0.3)

**Table 5 healthcare-12-02113-t005:** Variables correlate with neonatal outcomes (n = 300).

Variables	Correlation
	^a^ r
Age	r = 0.349 **
Hospitalization duration, (inpatients vs. outpatient)	r = 0.376 **
	^b^ r_pb
Intesive care unite (ICU) admission	0.560 *
COVID-19 clinical manifestations	0.430 **
Treatment settings	0.610 **
Maternal death	0.776 **

** *p* value < 0.01: * *p* value < 0.5; ^a^ Pearson’s correlation: ^b^ point-biserial correlation.

## Data Availability

The original contributions presented in the study are included in the article, further inquiries can be directed to the corresponding author/s.

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
