# Peer review of "Outcomes of COVID-19 in Pregnant Women: A Retrospective Analysis of 300 Cases in Jordan"

_healthcare, 2024, doi:10.3390/healthcare12212113_

Round 1
Reviewer 1 Report
Comments and Suggestions for Authors
See attachment

Minor editing suggested.
Author Response
|
Thank you so much for your valuable comments |
|
|
Reviewer: 1 |
|
|
The study’s aim was to look at impact of covid-19 on pregnant Jordanian women. They concluded from the results that COVID-19 in pregnant women, is associated with significant neonatal complications particularly in the third trimester. Below are a few suggestions that could help improve the content.
|
|
|
Line 31: Consider changing primary aim ‘is ’ to primary aim ‘was’
|
Noted and corrected , the document now read ‘ primary aim was” |
|
Line 32: Change includes to included
|
Noted and corrected |
|
Line 38: Change sever to severe
|
Noted and corrected |
|
Line 40: Change includes to included
|
Noted and corrected |
|
Line 45: Kindly remove the parenthesis around 33%
|
the parenthesis around 33% are removed |
|
Line 46 and 47. Kindly remove the parenthesis around the percentages
|
the parenthesis around the percentages are removed |
|
Line 86: Change Aims to aimed
|
Aims changed to aimed
|
|
Line 106: Change adopts to adopted
|
adopts changed to adopted
|
|
128: Kindly spell out the meaning of RCOG if applicable the first time it is mentioned in the text
|
RCOG is spilled out to The Royal College of Obstetricians and Gynecologists |
|
140 kindly spell out the meaning of HRCT if applicable the first time it is mentioned in the text
|
Noted and corrected |
|
Table 1 to 4: I will suggest that rather than have a separate column for percentages, you can put the percentages in parenthesis next to actual numbers.
|
Noted and corrected in all tables |
|
Figure 1: I see 3 comorbidities mentioned but the bars exceed 3. What do the others represent? |
Thank you very much for your valuable comment; we greatly appreciate this important feedback. In the previous version, we analyzed the frequency of comorbidities among the 300 participants, of which 55 reported having comorbidities. Based on your insightful note, we have reanalyzed the cases specifically among these 55 participants. Regarding your inquiry about the 3% mentioned in the earlier version of the document, this figure represented the following conditions within the 300 cases:
In our revised analysis, we included these conditions, which are now presented in the figure based on the analysis of the 55 participants who reported comorbidities.
|
|
Comments on the Quality of English Language --------------------------------------Minor editing suggested.
|
Editing and language checking was held by the senior author Tamara Darwish who is fluent in English and work in the UK |
Reviewer 2 Report
Comments and Suggestions for Authors
I have evaluated the submitted article very carefully. The mechanisms of action of COVID 19 as well as its relationship with different risk groups are still of interest to the medical world. The relationship between this viral infection and pregnancy is not fully understood. The work done by you contributes to the elucidation of these problems. Although I appreciate the work, I think it can be improved.
The inclusion criteria must be explained more clearly (were the cases consecutive?).
When talking about the abortions of these patients, it must be explained whether they were induced by the viral infection or had another cause. Also, the causes of death were associated with Covid19 or had other causes.
The text must be checked in terms of the font used.
Comments on the Quality of English Languageminor revision
Author Response
|
Thank you so much for your valuable comments |
|
|
Reviewer: 2 |
|
|
I have evaluated the submitted article very carefully. The mechanisms of action of COVID 19 as well as its relationship with different risk groups are still of interest to the medical world. The relationship between this viral infection and pregnancy is not fully understood. The work done by you contributes to the elucidation of these problems. Although I appreciate the work, I think it can be improved.
|
Thank you so much for this encouraging note |
|
The inclusion criteria must be explained more clearly (were the cases consecutive?).
|
Thank you so much for this valuable note, noted and corrected. Line numbers 179-180 now read’ The cases were collected consecutively by extracting data from the records of all eligible patients over the study period, ensuring no bias in case selection” |
|
When talking about the abortions of these patients, it must be explained whether they were induced by the viral infection or had another cause. Also, the causes of death were associated with Covid19 or had other causes.
|
Thank you for your valuable feedback. I would like to clarify that this study is based on a retrospective case series design, which inherently limits the ability to establish direct causal relationships, including whether the abortions or deaths were specifically induced by the viral infection or due to other causes.
As the data were collected from previously documented medical records, we relied on the information available. In many cases, detailed information on the precise cause of the abortion or death may not have been comprehensively documented, which is a limitation of the retrospective design. Nonetheless, we have provided the available information as thoroughly as possible and acknowledged this limitation in the discussion. |
|
The text must be checked in terms of the font used.
|
Editing and language checking was held by the senior author Tamara Darwish who is fluent in English and work in the UK |
|
Comments on the Quality of English Language minor revision |
Editing and language checking was held by the senior author Tamara Darwish who is fluent in English and work in the UK |
|
|
|
|
|
|
Reviewer 3 Report
Comments and Suggestions for Authors
It is an interesting work, with a consistent amount of studied cases that reveals the demographic, the therapeutic, the obstetrics and pediatric outcomes in a cohort of 300 COVID-19 positive pregnant Jordanian women. But the authors need to make serious corrections and changes to make it worth publication.
Abstract section
Row 37 explain the abbreviation PCR
Rephrase for clarity rows 37-39 – “Women infected with COVID-19 were categorized into four groups per RCOG Coronavirus (Covid-19), infection in pregnancy: Asymptomatic, Mild, Moderate and sever cases, all cases were treated as per RCOG management Protocol for Covid-19 infection in pregnancy.”
Explain the abbreviation RCOG
What does COVID-19 presentations mean? – COVID-19 clinical manifestations?
Most women were asymptomatic (61.7%), but (33%) had respiratory symptoms, (4.7 %) needed ICU admission, and (2.7%) resulted in maternal deaths. First trimester and second trimester miscarriages were recorded in (2.67%) and (3.67%) of cases respectively, while preterm labour occurred in 3.0% of pregnancies – please remove the parenthesis where here is no need for them or add the number of patients before the parenthesis – more recommended due to the small patient size, and correct labour into labor.
Introduction section
Please remove one of the modalities in which you wrote your references, the one with the superscript and leave the one according to journal’s specifications.
“However, a discernible gap remains in the literature—a comprehensive case series examining the maternal and neonatal outcomes within a substantial cohort of 300 COVID-19-positive pregnancies. ” – please rephrase, no verb in the second part of the phrase.
Row 83 – “COVID-19 presentations” same question as before? Does it refer to symptoms on admission?
Please check for errors and standardize the way you write COVID-19
“Hence, this study answers the following research questions” – please add the required punctuation after each question you stated.
“What are the correlated factors with COVID-19 among pregnant women who were diagnosed with COVID_19.” – please rephrase for clarity and add the correct punctuation, and use only COVID-19.
Do not use the number of the included patients in the introduction section, the number is a result of the inclusion and exclusion criteria applied so it will be stated in the results section.
Methods
Row 107 – “300”- Do not use the number of the included patients in the introduction section, the number is a result of the inclusion and exclusion criteria applied so it will be stated in the results section.
You use to many ways of writing COVID 19 – please select and use only one “COVID 19”/”COVID-19”/” COVID_19”/”Covid 19”???
“Proper history taking including, Onset, timing and severity of symptoms, contacts with infected patients, vaccinations and the presence of any comorbidities.” – please change the capital letter and add a verb to the sentence – row 118-119
Explain the abbreviations – BMI, O2, CBC, KFT, LFT, PT/INR, D-Dimer, CRP, LDH, CT
“chest x-ray and High-Resolution CT (HRCT)” – both were performed to the same patient? If not what were the criteria for performing CT and what for x-ray?
Row 126 add a “ ’ “ after ‘ informed consent.
“categorized into four groups per RCOG Coronavirus (Covid-19), infection in pregnancy (12).” – rephrase and change the punctuation form “.” Into “:”
“There were 4 management protocols per RCOG Coronavirus (Covid-19), infection in pregnancy:” – rephrase.
“Zocin 500 mg OD for 5 days, VIT C 500 mg tab BD, VIT D 1000 u OD and ZINC 25 mg OD for those with mild disease” “Admission, Rocephin 2g OD, Zocin 500 mg OD, Decadron IV 6 mg OD, Innohep 4500 IU OD, Vitamin C 500 MG Tab BD, Vitamin D D 1000 U OD, Zinc 25MG OD, Famotidine 40 MG OD” “Admission, Rocephin 2g OD, Zocin 500 mg OD, Decadron IV 6 mg OD, Innohep 4500 IU OD, VIT C 500 MG TAB BD, VIT D 1000 U OD, ZINC 25MG OD, FAMOTODINE 40 MG OD plus/minus t” explain the abbreviations and use the generic names.
“e: Antiviral Treatment (Remdesivir or Sancovir), Tocilizumab (Actemra) 153 and/or Hemoperfusion:” – rephrase, use the correct punctuation and do not capitalize first letter unless needed
“This study was approved by the Research Committee of the School of Medicine at Hashemite University. Keep in mind that the Ethics Committee waived the need for patient consent. To illustrate, the researcher applied for a waiver of patient consent because of the retrospective design utilized in this study. The waiver was granted based on the following criteria: (a) the research posed minimal or no risk to the women, (b) the data were anonymized before analysis to protect patients’ confidentiality, (c) obtaining individual consent was impractical due to the historical nature of the data, and (d) the study holds significant potential to contribute to understanding the maternal and neonatal 166 outcomes of pregnant Jordanian women with COVID-19 and to explore the factors correlated with the health-related outcomes for both the mothers and their infants.” – the paragraph has different font and extra spaces – please correct, and remove or rephrase “keep in mind” from here and from row 181.
Row 173 – “the head” – rephrase
State clearly your inclusion and exclusion criteria
Results section
Row 191 – “(demographics, obstetric history. COVID-19 related data, maternal Outcomes, and neonatal Outcomes)” – correct the punctuation, after the parenthesis no before
Row 194 – “Age” needs to be with capital letter??? “maternal Outcomes” ??? “(inpatients Vs outpatient)” , neonatal Outcomes - lower case
ICU – explain?
“Outcomes sheet that has two variables only; mode of delivery, and neonatal outcomes.” – correct the punctuation.
Please rephrase and correct the whole 2.5. Measures chapter – use correct and an homogenous verb form.
2.6. Data Analysis - correct because there are different fonts
“The demographics, obstetrics framework and comorbidities among Jordanian pregnant women with COVID-19” – if this is a subtitle please mark it as being one if it is a phrase make it to be one.
Row 220 – “269(89.7%) “ add a space
Table 1 – “Obstraric History” please correct; delete extra space “1 -3 times”; explain the abbreviation CS; “pregnancy complications” capitalize first letter and write with lower case the words NONE ANEMIA and explain the abbreviation GDM GHTN; Anemia And Vaginal Bleeding are written like this but Age group like this??
Row 229 “sample (55 Patients)” correct the upper case; Type Two DM Multiple comorbidities ? lower upper case; rephrase the whole sentence for clarity.
Correct figure 1 – looks sloppy
Row 235 – “182(61.7%) ” add space
“Patients presented with respiratory symptoms” - patients with lower case; explain GI abbreviation
“The disease severity was categorized into four categories as previously mentioned: Asymptomatic, mild, moderate, and severe, (53.3%) of 239 the total sample were asymptomatic (160 cases), 63 (21%) had mild symptoms, 50 (6.7%) 240 and 27 (9.0%) presented with moderate and severe symptoms respectively.” – rephrase, use the correct punctuation and lower case and the parenthesis must be used when needed.
“34(11.3%) received protocol C (Patients)” – add extra space and why does Patients with capitalized letter is between parenthesis? Please correct
Correct table 2 – different fonts, different ways to write GI symptoms
Please add to table 1 and 2, p values or CI as you stated “means with standard deviations or medians with interquartile ranges, depending on the distribution”
Correct Table 3 – without mean and SD “Hospital stays days; Mean (SD);[Range]: 1.8, (2.48); [0-15]” - you can add those as table columns
“Treatment options” ???? – in table 3 – no options described
Correct table 4 – written only with upper case, no explained abbreviations
All the tables need proper formatting with first column aligned to the left and the other columns centered, no statistics but descriptive please add p values median interquartile range.
Add to table 5 what statistics test you used for correlation
Discussion section
“Ullah et al. 282 (2020)(15) 15 and Akhtar et al. (2020)(16) 16,” – correct this remove the year from the parenthesis; please do that in all the dissection section.
“The United States”; “However, “ - lower case
Pre-eclampsia – correct it
“in France yielded (2%) rate” ; “outcomes where (87.3%) were born alive while (4.0%) were stillbirths “ - remove the parenthesis
“In Montreal,Quebec, women with asymptomatic Covid- 19 ” – add needed spaces and correct COVID-19
At the end of the discussions section state study limitations and its strengths.
PLEASE REWRITE THE DISSCUTION SECTION AND MAKE IT MORE DOCUMENTED FROM THE EXISTING BODY OF LITERATURE.
The conclusion section needs to contain conclusions of your actual results not general statements – please correct
“Declarations 377 Ethics approval and consent to participate: This study was approved by the Re- 378 search Committee of the School of Medicine at Hashemite University/Jordan. 379 Consent for publication:Not applicable 380 Availability of data and materials: No, I don't have any research data outside the sub- 381 mitted manuscript file. 382 Competing interests:The authors declare that they have no competing interests. 383 Funding:No funding 384 Acknowledgments: we acknowledge the Hashemite university and Queen Alia Hospital for 385 the logistic support.” – different fonts, some need spacing – please correct

Moderate English revision is needed.
Author Response
|
Thank you so much for your valuable comments |
|
|
|
|
|
Reviewer: 3 |
|
|
It is an interesting work, with a consistent amount of studied cases that reveals the demographic, the therapeutic, the obstetrics and pediatric outcomes in a cohort of 300 COVID-19 positive pregnant Jordanian women. But the authors need to make serious corrections and changes to make it worth publication. |
Thank you so much for your time and effort. |
|
Abstract section
|
|
|
Row 37 explain the abbreviation PCR
|
Noted and corrected. Line numbers 36-37 now read ‘Antigen Test (RAT) + long Polymerase Chain Reaction (PCR) test used to detect SARS-CoV-2 by amplifying viral RNA from patient samples” |
|
Rephrase for clarity rows 37-39 – “Women infected with COVID-19 were categorized into four groups per RCOG Coronavirus (Covid-19), infection in pregnancy: Asymptomatic, Mild, Moderate and sever cases, all cases were treated as per RCOG management Protocol for Covid-19 infection in pregnancy.”
|
Thank you so much, noted and corrected, and the document now reads ‘Women infected with COVID-19 were categorized into four groups according to the Royal College of Obstetricians and Gynecologists guidelines for COVID-19 infection in pregnancy: asymptomatic, mild, moderate, and severe cases. All cases were managed following the RCOG protocol for COVID-19 in pregnancy” |
|
Explain the abbreviation RCOG |
Noted and corrected, it now reads ‘the Royal College of Obstetricians and Gynecologists (RCOG) |
|
What does COVID-19 presentations mean? – COVID-19 clinical manifestations? |
Thank you so much, the document now reads ‘COVID-19 clinical manifestations” |
|
Most women were asymptomatic (61.7%), but (33%) had respiratory symptoms, (4.7 %) needed ICU admission, and (2.7%) resulted in maternal deaths. First trimester and second trimester miscarriages were recorded in (2.67%) and (3.67%) of cases respectively, while preterm labour occurred in 3.0% of pregnancies – please remove the parenthesis where here is no need for them or add the number of patients before the parenthesis – more recommended due to the small patient size, and correct labour into labor. |
Thank you so much for this note, The parenthesis were removed; labour is not read labor.
|
|
Introduction section |
|
|
Please remove one of the modalities in which you wrote your references, the one with the superscript and leave the one according to journal’s specifications. |
Thank you for your insight, we removed the superscripts. |
|
“However, a discernible gap remains in the literature—a comprehensive case series examining the maternal and neonatal outcomes within a substantial cohort of 300 COVID-19-positive pregnancies. ” – please rephrase, no verb in the second part of the phrase. |
Thank you so much, noted and the document now reads ‘However, there remains a discernible gap in the literature: a comprehensive case series examining maternal and neonatal outcomes in a substantial cohort of 300 COVID-19-positive pregnancies has yet to be conducted. |
|
Row 83 – “COVID-19 presentations” same question as before? Does it refer to symptoms on admission? |
Now changed to ‘clinical manifestations” |
|
Please check for errors and standardize the way you write COVID-19 |
Thank you so much for this note, we have checked it and now it is written “COVID-19” throughout the whole document |
|
“Hence, this study answers the following research questions” – please add the required punctuation after each question you stated. |
Noted and corrected and now a question mark is added after each question. |
|
“What are the correlated factors with COVID-19 among pregnant women who were diagnosed with COVID_19.” – please rephrase for clarity and add the correct punctuation, and use only COVID-19. |
Noted and rephrased and the question now reads ‘What factors are correlated with COVID-19 among pregnant women diagnosed with COVID-19?” |
|
Do not use the number of the included patients in the introduction section, the number is a result of the inclusion and exclusion criteria applied so it will be stated in the results section. |
Thank you for this insight, the number now removed from the research questions. Hence, research question number 2 now reads ‘What are the COVID-19 related characteristics among pregnant women in Jordan?’ |
|
Methods |
|
|
Row 107 – “300”- Do not use the number of the included patients in the introduction section, the number is a result of the inclusion and exclusion criteria applied so it will be stated in the results section. |
Noted and corrected ‘The first two lines now read “This research adopted a retrospective case series design, meticulously analyzing the maternal and neonatal outcomes of COVID-19-positive pregnancies” |
|
You use to many ways of writing COVID 19 – please select and use only one “COVID 19”/”COVID-19”/” COVID_19”/”Covid 19”??? |
Thank you so much for this insight, it was checked throughout the document and now it has written in the following format “COVID-19)” |
|
“Proper history taking including, Onset, timing and severity of symptoms, contacts with infected patients, vaccinations and the presence of any comorbidities.” – please change the capital letter and add a verb to the sentence – row 118-119 |
Thank you so much, the document now reads “Proper history taking includes the onset, timing, and severity of symptoms, contacts with infected patients, vaccinations, and the presence of any comorbidities” |
|
Explain the abbreviations – BMI, O2, CBC, KFT, LFT, PT/INR, D-Dimer, CRP, LDH, CT |
Noted and clarified, the document now reads “. Laboratory tests including Complete Blood Count (CBC), Kidney Function Test (KFT), Liver Function Test (LFT), Prothrombin Time/International Normalized Ratio (PT/INR), protein fragment present in the blood after a blood clot dissolves, used to rule out thrombotic disorders (D-Dimer), C-Reactive Protein (CRP), Lactate Dehydrogenase (LDH) and Ferritin were undertaken for all patients, chest x-ray and High-Resolution Computed Tomography (CT) and High-Resolution Computed Tomography (HRCT)” |
|
“chest x-ray and High-Resolution CT (HRCT)” – both were performed to the same patient? If not what were the criteria for performing CT and what for x-ray? |
The chest X-ray was used for initial evaluation, while the HRCT was conducted based on specific criteria, such as the need for detailed imaging due to abnormal findings on the X-ray or clinical symptoms indicating more complex lung conditions |
|
Row 126 add a “ ’ “ after ‘ informed consent. |
Noted and corrected |
|
categorized into four groups per RCOG Coronavirus (Covid-19), infection in pregnancy (12).” – rephrase and change the punctuation form “.” Into “:” |
Noted and the document now reads =Following this comprehensive assessment, women infected with COVID-19 were categorized into four groups according to the RCOG guidelines for COVID-19 infection in pregnancy” |
|
There were 4 management protocols per RCOG Coronavirus (Covid-19), infection in pregnancy:” – rephrase. |
Noted and the document now reads “There were four management protocols according to the RCOG guidelines for COVID-19), infection in pregnancy” |
|
“Zocin 500 mg OD for 5 days, VIT C 500 mg tab BD, VIT D 1000 u OD and ZINC 25 mg OD for those with mild disease” “Admission, Rocephin 2g OD, Zocin 500 mg OD, Decadron IV 6 mg OD, Innohep 4500 IU OD, Vitamin C 500 MG Tab BD, Vitamin D D 1000 U OD, Zinc 25MG OD, Famotidine 40 MG OD” “Admission, Rocephin 2g OD, Zocin 500 mg OD, Decadron IV 6 mg OD, Innohep 4500 IU OD, VIT C 500 MG TAB BD, VIT D 1000 U OD, ZINC 25MG OD, FAMOTODINE 40 MG OD plus/minus t” explain the abbreviations and use the generic names. |
Noted and corrected and the document now reads “ B Zocin /Piperacillin/Tazobactam 500 mg /an antibiotic given once daily (OD) for five days; VIT C/ Vitamin C 500 mg tablet twice daily (BD); VIT D/ Vitamin D 1000 u once daily (OD); and ZINC 25 mg once daily (OD). C Admission, Rocephin / Ceftriaxone, an antibiotic given at 2 grams once daily (OD). , Zocin /Piperacillin/Tazobactam 500 mg /an antibiotic given once daily (OD) for five days; Decadron / Dexamethasone, a corticosteroid given intravenously (IV) at 6 mg once daily (OD).; Innohep/Tinzaparin 4500 IU , a low molecular weight heparin (anticoagulant) given at 4500 IU once daily (OD).; VIT C/ Vitamin C 500 mg tablet twice daily (BD); VIT D/ Vitamin D 1000 u once daily (OD); and ZINC 25 mg once daily (OD)., Famotidine 40 mg OD: Famotidine, an acid-reducing medication given at 40 mg once daily (OD)” |
|
“e: Antiviral Treatment (Remdesivir or Sancovir), Tocilizumab (Actemra) 153 and/or Hemoperfusion:” – rephrase, use the correct punctuation and do not capitalize first letter unless needed |
Noted and corrected “Antiviral treatment (remdesivir or sancovir), tocilizumab (Actemra), and/or hemoperfusion” |
|
“This study was approved by the Research Committee of the School of Medicine at Hashemite University. Keep in mind that the Ethics Committee waived the need for patient consent. To illustrate, the researcher applied for a waiver of patient consent because of the retrospective design utilized in this study. The waiver was granted based on the following criteria: (a) the research posed minimal or no risk to the women, (b) the data were anonymized before analysis to protect patients’ confidentiality, (c) obtaining individual consent was impractical due to the historical nature of the data, and (d) the study holds significant potential to contribute to understanding the maternal and neonatal 166 outcomes of pregnant Jordanian women with COVID-19 and to explore the factors correlated with the health-related outcomes for both the mothers and their infants.” – the paragraph has different font and extra spaces – please correct, and remove or rephrase “keep in mind” from here and from row 181. |
Noted and corrected |
|
Row 173 – “the head” – rephrase |
The head changed to the manager |
|
State clearly your inclusion and exclusion criteria |
The cases were collected consecutively by extracting data from the records of all eligible patients over the study period, ensuring no bias in case selection. The inclusion criteria were pregnant women diagnosed with COVID-19, confirmed through PCR testing. The exclusion criteria were women who suffer from chronic obstructive pulmonary disease (COPD) and lung cancer. |
|
Results section |
|
|
Row 191 – “(demographics, obstetric history. COVID-19 related data, maternal Outcomes, and neonatal Outcomes)” – correct the punctuation, after the parenthesis no before |
Noted and corrected |
|
Row 194 – “Age” needs to be with capital letter??? “maternal Outcomes” ??? “(inpatients Vs outpatient)” , neonatal Outcomes - lower case |
Noted and corrected |
|
ICU – explain? |
Noted and corrected |
|
“Outcomes sheet that has two variables only; mode of delivery, and neonatal outcomes.” – correct the punctuation. |
Noted and corrected |
|
Please rephrase and correct the whole 2.5. Measures chapter – use correct and an homogenous verb form. |
Noted and corrected. The section now reads “The current study utilized five data sheets to extract the required data. These sheets were developed in advance by the researchers based on existing literature and systemati-cally captured data, including demographics, obstetric history, COVID-19-related data, maternal outcomes, and neonatal outcomes. The following paragraphs provide a brief discussion of each data sheet.
The demographic sheet includes two variables: age and smoking habits. The obstetric history sheet comprises ten variables, including obstetric history, previous mode of deliv-ery, gestational age at diagnosis, comorbidities, and pregnancy complications. The COVID-19-related data sheet covers four variables: clinical presentation, O2 saturation, disease severity, and treatment protocol. The maternal outcomes sheet consists of the fol-lowing variables: duration of hospitalization, treatment options (inpatient vs. outpatient), ICU admission, and maternal death. Lastly, the pregnancy and neonatal outcomes sheet contains two variables: mode of delivery and neonatal outcomes”.
|
|
2.6. Data Analysis - correct because there are different fonts |
Noted and corrected |
|
“The demographics, obstetrics framework and comorbidities among Jordanian pregnant women with COVID-19” – if this is a subtitle please mark it as being one if it is a phrase make it to be one. |
Deleted |
|
Row 220 – “269(89.7%) “ add a space |
Noted |
|
Table 1 – “Obstraric History” please correct; delete extra space “1 -3 times”; explain the abbreviation CS; “pregnancy complications” capitalize first letter and write with lower case the words NONE ANEMIA and explain the abbreviation GDM GHTN; Anemia And Vaginal Bleeding are written like this but Age group like this?? |
Noted and corrected |
|
Row 229 “sample (55 Patients)” correct the upper case; Type Two DM Multiple comorbidities ? lower upper case; rephrase the whole sentence for clarity. |
Noted and corrected |
|
Correct figure 1 – looks sloppy |
Noted and corrected. In the previous version, we analyzed the frequency of comorbidities among the 300 participants, of which 55 reported having comorbidities. Based on your insightful note, we have reanalyzed the cases specifically among these 55 participants. The figure has now been corrected. |
|
Row 235 – “182(61.7%) ” add space |
Corrected |
|
“Patients presented with respiratory symptoms” - patients with lower case; explain GI abbreviation |
Noted |
|
“The disease severity was categorized into four categories as previously mentioned: Asymptomatic, mild, moderate, and severe, (53.3%) of 239 the total sample were asymptomatic (160 cases), 63 (21%) had mild symptoms, 50 (6.7%) 240 and 27 (9.0%) presented with moderate and severe symptoms respectively.” – rephrase, use the correct punctuation and lower case and the parenthesis must be used when needed. |
Noted and the document now reads “The severity of the disease was classified into four categories, as previously mentioned: asymptomatic, mild, moderate, and severe. Among the total sample, 53.3% were asymp-tomatic (160 cases), 21% had mild symptoms (63 cases), 6.7% presented with moderate symptoms (50 cases), and 9.0% exhibited severe symptoms (27 cases)”. |
|
34(11.3%) received protocol C (Patients)” – add extra space and why does Patients with capitalized letter is between parenthesis? Please correct |
Noted |
|
Correct table 2 – different fonts, different ways to write GI symptoms |
Noted |
|
Please add to table 1 and 2, p values or CI as you stated “means with standard deviations or medians with interquartile ranges, depending on the distribution” |
Thank you for your valuable feedback. Table 1 and 2 represent frequency and descriptive data, and as such, we did not conduct statistical tests like t-tests or ANOVA that would produce p-values or confidence intervals. As you are aware, frequency and descriptive analyses typically provide results without these values. Instead, we used correlation coefficients to identify relationships between the variables. We apologize for any confusion and appreciate your understanding. |
|
Correct Table 3 – without mean and SD “Hospital stays days; Mean (SD);[Range]: 1.8, (2.48); [0-15]” - you can add those as table columns |
Noted and corrected |
|
“Treatment options” ???? – in table 3 – no options described |
Thank you for your valuable feedback. In Table 3, the "Treatment options" refer to the inpatient and outpatient care settings, which were used to categorize the treatment approaches. We aimed to classify the types of care provided, rather than detailing specific medical treatments. However, we recognize that this may not have been clear, and revised the table and now the table reads “Ttreatment settings” . |
|
Correct table 4 – written only with upper case, no explained abbreviations |
Noted and corrected |
|
All the tables need proper formatting with first column aligned to the left and the other columns centered, no statistics but descriptive please add p values median interquartile range. |
Thank you for your valuable feedback regarding the formatting of the tables and the statistical details. I appreciate your suggestions for enhancing clarity and presentation.
I would like to clarify that only the age and the Hospital stays days data were collected in a continuous manner, and we provided the median and IQR, the other variables were categorical (e.g., grouped age ranges). As such, measures like median and interquartile range are not applicable for these categorical data. Instead, our focus has been placed on presenting descriptive statistics, such as percentages and counts, to accurately reflect the data.
Thank you once again for your insightful comments. I hope this clarification addresses your concerns, and I look forward to any further suggestions you may have. |
|
Add to table 5 what statistics test you used for correlation |
Noted and corrected |
|
Discussion section |
|
|
Ullah et al. 282 (2020)(15) 15 and Akhtar et al. (2020)(16) 16,” – correct this remove the year from the parenthesis; please do that in all the dissection section. |
Noted and corrected |
|
The United States”; “However, “ - lower case |
Noted and corrected to “the United State “ |
|
Pre-eclampsia – correct it |
Noted and corrected to “Preeclampsia” |
|
“in France yielded (2%) rate” ; “outcomes where (87.3%) were born alive while (4.0%) were stillbirths “ - remove the parenthesis |
Noted and corrected |
|
In Montreal,Quebec, women with asymptomatic Covid- 19 ” – add needed spaces and correct COVID-19 |
Noted and corrected |
|
At the end of the discussions section state study limitations and its strengths. |
Noted and corrected |
|
The conclusion section
|
|
|
needs to contain conclusions of your actual results not general statements – please correct |
|
|
“Declarations 377 Ethics approval and consent to participate: This study was approved by the Re- 378 search Committee of the School of Medicine at Hashemite University/Jordan. 379 Consent for publication:Not applicable 380 Availability of data and materials: No, I don't have any research data outside the sub- 381 mitted manuscript file. 382 Competing interests:The authors declare that they have no competing interests. 383 Funding:No funding 384 Acknowledgments: we acknowledge the Hashemite university and Queen Alia Hospital for 385 the logistic support.” – different fonts, some need spacing – please correct |
Noted and corrected |
|
Moderate English revision is needed. |
Editing and language checking was held by the senior author Tamara Darwish who is fluent in English and work in the UK |
|
|
|
Round 2
Reviewer 3 Report
Comments and Suggestions for Authors
I am satisfied with the corrections.